# Proposal for a Battery to Evaluate Functional Capacity in Older Adults with Disabilities

**DOI:** 10.3390/s25061813

**Published:** 2025-03-14

**Authors:** Josu Ascondo, Iñaki Llodio, Bingen Marcos-Rivero, Cristina Granados, Sheila Romero, Aitor Iturricastillo, Javier Yanci

**Affiliations:** 1Society, Sports and Physical Exercise Research Group (GIKAFIT), Physical Education and Sport Department, Faculty of Education and Sport, University of the Basque Country, UPV/EHU, 01007 Vitoria-Gasteiz, Spain; josu.ascondo@ehu.eus (J.A.); bingen.marcos@ehu.eus (B.M.-R.); cristina.granados@ehu.eus (C.G.); javier.yanci@ehu.eus (J.Y.); 2Research Group in Physical Activity, Physical Exercise and Sport (AKTIBOki), Physical Education and Sport Department, Faculty of Education and Sport, University of the Basque Country, UPV/EHU, 01007 Vitoria-Gasteiz, Spain; 3Physical Activity, Exercise, and Health Group, Bioaraba Health Research Institute, 01007 Vitoria-Gasteiz, Spain; 4Disability Research Department, GaituzSport Foundation, 48003 Bilbao, Spain; romero.sheila@gaituzsport.eus

**Keywords:** physical condition, functionality, validity, repeatability

## Abstract

The purpose was to design and validate a battery of physical tests, called EFEPD-1.0, adapted to assess functionality in people with disabilities. In addition, we sought to analyze the validity and reliability of this battery both for the total group and differentiated by sex. A total of 43 adults with disabilities (32 women and 11 men) participated (57.11 ± 10.12 years). The battery was composed of five blocks of functionality: neuromuscular, combined actions, acceleration, balance, and cardiovascular. The neuromuscular functionality was measured by the vertical and horizontal jump test using the optical system (Opto Jump Next^®^, Microgate, Bolzano, Italy) as well as the Hand Grip (HG) test using a (5030J1, Jamar^®^, Sammons Preston, Inc, Nottinghamshire, UK) hand dynamometer. The combined actions and balance functionality were assessed with the Time Up and Go (TUG) test, the 30 s Chair Stand (30CTS) test, and the One-Leg Stance (OLS) test measured by a manual stopwatch (HS-80TW-1EF, Casio^®^, Tokyo, Japan). The acceleration functionality was evaluated through 20 m sprints and the 505 change of direction (COD505) test, using the (Microgate, Witty^®^, Bolzano, Italy) photocell system. The cardiovascular functionality was evaluated with the Six-Minute Walking Test (6MWT), where heart rate was monitored using the (Polar Team Sport System^®^, Polar Electro Oy, Kempele, Finland), and additional walking mechanics were recorded with Stryd (Stryd Everest 12 Firmware 1.18 Software 3, Stryd Inc., Boulder, CO, USA). The results showed that the intraclass correlation coefficients (ICCs) ranged from moderate to almost perfect (ICC = 0.65–0.98) between test repetitions. Some tests could significantly differentiate (*p* < 0.05) men and women, highlighting better neuromuscular capacity in men and better balance in women. The correlations between tests showed significant convergent validity. The Evaluation of Functionality in the Disabled Population (EFEPD-1.0) battery not only consistently measures functional capacities in people with disabilities, but it can also discriminate between different subgroups within this population.

## 1. Introduction

Disability, understood as a limitation in bodily functionality or structure that results in difficulties in carrying out daily tasks and participating fully in society [1], is a global reality that affects a large number of people. According to the World Health Organization (WHO), approximately 15% of the world’s population lives with some type of disability [2]. Although the impact of disability can vary considerably across continents and nations, in the European Union, the average prevalence of people with disabilities is 24% of the total population, reaching figures of up to 39.5% in some countries [3]. Although the percentage of the population with disabilities can also vary according to the type of disability, age, gender, and other factors, it seems clear that it has a high incidence in the world population [4].

It has been described that people with disabilities have reduced health due to the impact of the disability itself [5] and to their lifestyle [6]. Different studies have shown that people with disabilities have worse physical [7] and psychological [8] health compared to the non-disabled population. While physical activity (PA) has been described as a key factor for improving health and well-being in the population with disabilities [9], a lack of PA or a high level of sedentary lifestyle is a common problem in this population [6]. In this sense, there is evidence that people with disabilities have a lower level of PA practice and a longer time of physical inactivity than people without disabilities [5,6]. Considering that a sedentary lifestyle can have negative health consequences, including an increased risk of cardiovascular disease and diabetes, a reduction in quality of life [10], and even psychological and social alterations [11], it seems important to promote PA in people with disabilities as a way to improve their health and well-being.

One of the barriers to the practice of PA that people with disabilities face is their own disability and a low level of physical fitness (PF) [12]. PF, understood as an individual’s ability to perform the physical activities of daily living with vigor and efficiency, has been defined as a biological marker of health status, as well as a powerful predictor of longevity and quality of life [13]. Along these lines, different studies have focused on analyzing the relationship between the components of PF and the practice of PA, reporting that PA seems to be a good tool to improve these markers and therefore improve health [14,15]. Therefore, improving the PF of people with disabilities may be one of the priorities to increase PA practice and improve health.

Although the analysis of PF in people with disabilities has been carried out in multiple studies, in both sports performance [16] and health [17], Alcántara-Cordero et al. [18] highlight the lack of standardized tests and batteries to assess PF in people with disabilities. Thus, in the scientific literature, different batteries of tests have been used to determine the PF of people with disabilities, but there is no consensus between the functional test batteries or the tests that make them up [19]. Moreover, many of them are adaptations of non-disabled population tests [20], with little evaluation of validity and reliability [19] for applying in people with disabilities. For example, the EUROFIT battery has been modified to adapt it to people with intellectual disabilities and has become one of the most widely used batteries in this population, but due to the difficulty in understanding the instructions and performing some of the proposed exercises, some studies have reported limitations in the applicability of this type of battery in that population [18]. Similarly, the Assessing Levels of Physical Activity (ALPHA-Fit) batteries, modified to assess the physical condition of children and adolescents with intellectual disabilities, and the Brockport Physical Fitness Test (BPFT) battery used in people with disabilities to assess functional capacity and health-related physical performance have significant limitations [21,22]. These limitations are mainly related to the study sample and to the applicability of the tests. In fact, the samples of the validation studies of these batteries are made up of people with a single type of disability [18,22], or the age range of the samples is specific to children or older adults [22]. Moreover, some of the suggested tests have not been modified and, therefore, they cannot be implemented adequately [18]. When the tests have been adapted to the population with disabilities, the psychometric tests of validity and reliability have not been sufficiently evaluated [19]. In this sense, it seems necessary to design a battery of tests for a wide spectrum of people with disabilities or limited functionality and to determine their validity and reliability.

Therefore, the objectives of the present study were (1) to design a battery of tests to measure PF adapted to older adults with disabilities or functional limitations and (2) to analyze the intra-session repeatability, and the convergent and discriminant validity of the proposed battery.

## 2. Materials and Methods

### 2.1. Participants

This study involved 43 adults (57.11 ± 10.12 years; range: 28–75), with some disability or limitation of functionality; 31 of them were women (56.43 ± 9.64 years; range: 28–75) and 11 were men (59.09 ± 11.69 years; range: 32–71). A total of 51.2% of the participants had a functional limitation as a result of having suffered from cancer (CD), 27.9% had a physical disability (FD), 16.3% had an intellectual disability (ID), and 4.7% did not know/or want to answer what type of disability or limitation of functionality they had. The inclusion criteria for participating in this study were the following: (1) having an officially diagnosed functional limitation, pathology, or disability, (2) being regular users of some type of supervised and organized physical exercise or sport program, and (3) taking the complete battery of tests. Before starting the research, this project was approved by the Ethics Committee for Research with Human Beings (CEISH, code M10_2020_244) of the University of the Basque Country (UPV/EHU) and followed the requirements established in the Declaration of Helsinki in 2013.

### 2.2. Procedure

Before starting with the Evaluation of Functionality in the Disabled Population (EFEPD-1.0) battery, all the participants performed a standardized warm-up that consisted of 3 min of low-intensity movement, 4 progressive sprints of 20 m, and 4 sprints of 10 m with 2 changes of direction in each of them. The EFEPD-1.0 test battery was subdivided into 5 blocks of functionality and followed this order: neuromuscular functionality, functionality in combined action, functionality in acceleration, functionality in balance, and cardiovascular functionality, for which flexible sensors were used to record each test [23]. Each participant attended a single session to perform the battery. The sessions were carried out in small groups of between 4 and 8 people. The rest between tests was 120 s, enough time to give the pertinent explanations and provide a practical example. The participants performed two repetitions (R1 and R2) in each of the tests, except for the cardiovascular function test, in which a single attempt was made. In all the test sessions, the participants were informed that in each of the tests they should exert themselves to the maximum as long as they did so safely.

### 2.3. Measurement

#### 2.3.1. Neuromuscular Functionality

Vertical jump: In order to measure the functionality of the participants’ vertical jump, the bipodal (CMJ) and unipodal countermovement jump tests were used, both with the right leg (CMJ_Right_) and with the left leg (CMJ_Left_), previously described in the literature for people with disabilities [24]. In all cases, the participants were placed in an upright position in order to perform the bipodal or unipodal jumping protocol with established countermovement. The participants freely chose the degree of flexion [25] and their hands had to stay on their hips throughout the jump [26]. Each participant performed two maximum attempts of each type of jump, with 30 s of rest between the trials. In all the attempts, verbal instructions and stimuli were provided for better performance. An optical data collection system (Opto Jump Next^®^, Microgate, Bolzano, Italy) was used to measure the height of the jump.

Horizontal jump: To determine the functionality of the horizontal jump, the Standing Broad Jump (SBJ) test, previously described for people with disabilities, was used [27]. The participants positioned themselves behind a set line and performed a jump with both legs together on the horizontal axis as far as they could. In the execution of the test, the participants were allowed to use their arms to increase the jumping distance. All the participants performed two attempts with 30 s of rest between the trials. In all the attempts, verbal instructions and stimuli were provided for better performance. The jumped distance was measured from the starting line to the support of the furthest heel [28].

Upper extremity strength: In order to determine the isometric strength of the upper limbs, the Hand Grip (HG) test was used, previously used in people with disabilities [18]. The participants performed the test sitting in a chair with their right arm fully extended and without touching the chair [29], having to exert the highest isometric grip force for 5 s. A portable hydraulic hand dynamometer (5030J1, Jamar^®^, Sammons Preston, Inc., Nottinghamshire, UK) was used to measure the HG strength. Each participant performed two attempts with each of the hands (right and left) with 10 s of rest between each trial.

#### 2.3.2. Functionality in Combined Actions

Displacement from a sitting position: In order to determine functionality in standing, moving, turning, and sitting, the Time Up and Go (TUG) test was used, previously used in the population with disabilities [18]. The participants began the test sitting in a chair, with their feet in contact with the ground and their hip and knee joints bent to about 90°. At the voice saying “Go”, the participants got up and moved as quickly as possible to a cone located 3 m away; they surrounded it and returned to the starting position. Each participant performed two attempts with a 30 s break between each trial [18]. The time(s) taken to perform the test was measured using a manual stopwatch (HS-80TW-1EF, Casio^®^, Tokyo, Japan).

Lower limb functionality: In order to determine the functionality of the lower limbs, the 30 s Chair Stand test (30CTS) was used [30]. A fixed-height (45 cm) armless chair was used for the test, in which the participants had to sit in the center without leaning on the backrest, with their hands on opposite shoulders and their feet in contact with the floor [31]. Starting from the mentioned position, upon hearing the “Go” signal, the participants had to get up and sit down to complete the greatest number of cycles for 30 s. The number of cycles completed during the test was counted. The test was repeated twice with a rest time of 30 s.

#### 2.3.3. Functionality in Acceleration

Straight-line acceleration: The 20 m acceleration test, previously used in the population with disabilities, was used to measure functionality in straight-line acceleration [32]. The participants stood at the starting line and covered, in the shortest possible time, a distance of 20 m in a straight line. The starting line was set 0.5 m behind the first photocell (Microgate, Witty^®^, Bolzano, Italy) and the stopwatch was automatically activated when the participants passed the first gate at the 0.0 m mark [29]. Each participant performed two maximum accelerations with a rest of 120 s between each trial [33]. In all the attempts, verbal instructions and stimuli were provided for the participants to perform the accelerations at maximum intensity. The split times of 0–10 m, 0–20 m, and 10–20 m were recorded.

Acceleration with change of direction: In order to determine the functionality of acceleration with change of direction, the 505 change of direction test (COD505) was used. This test has been previously used in the population with disabilities [34]. The participants, placed on the starting line 10 m from the photocell (Microgate, Witty^®^, Bolzano, Italy), sprinted forward to a line 5 m ahead of the photocell, pivoted 180°, and accelerated for another 5 m to cross the gate again. The time was activated and stopped automatically when the participants crossed the photocell gate. Each of the participants completed two trials with a 120s rest period between them. 

#### 2.3.4. Functionality in Balance

Static balance: In order to determine balance functionality, the One-Leg Stance (OLS) test was used [35]. The participants were placed in an upright position on one leg and with their arms crossed over their chest, with both the right leg (OLS_Right_) and the left leg (OLS_Left_). The timer started when the participant lifted their foot off the ground. The test ended when the participant performed any of the following actions: 1. uncrossing or using the arms to maintain balance; 2. touching the ground with the foot raised; 3. moving the foot that supported the weight of the body; or 4. exceeding the maximum duration of 20 s. Two trials were performed for each leg (OLS_Right_ and OLS_Left_) with 30 s of rest between the trials [28].

#### 2.3.5. Cardiovascular Functionality

Six-Minute Walking Test (6MWT): The 6MWT was used to measure cardiovascular functionality [36]. The test was conducted in pairs, with the aim of covering the maximum possible distance within a marked rectangle using cones over a duration of 6 min [37]. The distance per lap was 50 m, and the total distance covered was recorded by a lap-counting system. Additionally, heart rate monitors (Polar Team Sport System^®^, Polar Electro Oy, Kempele, Finland) were used throughout the test to monitor the cardiac response (maximum heart rate [HRmax] and average heart rate [HRavg]) [38]. Stryd devices (Stryd Everest 12 Firmware 1.18 Software 3, Stryd Inc., Boulder, CO, USA) were also employed to record mechanical walking variables, such as the relative maximum power (Pmax_rel_), relative average power (Pavg_rel_), relative minimum power (Pmin_rel_), absolute maximum power (Pmax_abs_), absolute average power (Pavg_abs_), absolute minimum power (Pmin_abs_), cadence, stride length, pace, ground contact time (GCT), and total distance covered (Dist.Stryd) [39]. Immediately following the test, the participants reported their muscular (RPEmus) and respiratory (RPEres) values of subjective perception of effort using the Foster 0–10 scale [40], and the tympanic temperature was measured (ThermoScan^®^ IRT 4520 5, Braun GmbH, Kronberg, Germany) [41].

### 2.4. Statistical Analysis

The results are presented as the mean and standard deviation (SD). The Shapiro–Wilk and Levene tests were conducted to assess the normality and homoscedasticity of the data, respectively. The intra-session repeatability of the tests was evaluated using the intra-subject coefficient of variation (CV), the T-Student test for related samples, the Wilcoxon signed-rank test for non-parametric variables, the calculation of the magnitudes of differences through Cohen’s effect size (ES) [42] or the rank-biserial correlation (r_b_) for non-parametric variables, the Pearson (r) or Spearman (Rho) correlation coefficients, and the ICC (mixed-model, two-way, single measures, absolute agreement). To determine the differences between men and women, the independent samples T-Student test or the Mann–Whitney test for non-parametric variables were used. The calculation of the magnitudes of differences was analyzed through the ES or the probability of superiority (PS) for non-parametric variables [43]. For correlations among variables from the tests belonging to the blocks of neuromuscular functionality, combined action functionality, acceleration functionality, balance functionality, and the variables from the 6MWT, the Pearson (r) or Spearman (Rho) correlation coefficients were used. The qualitative interpretation of the ES values was as follows: d < 0.25, *trivial*; d = 0.25–0.50, *small*; d = 0.50–1.00, *moderate*; and d > 1.00, *large* [42]. The qualitative interpretation for the rank-biserial correlation (r_b_) values used the following scale: *trivial*, r_b_ < 0.10; *small*, r_b_ = 0.10–0.29; *moderate*, r_b_ = 0.30–0.49; and *large*, r_b_ > 0.50 [44]. The interpretation of the PS values used the following scale: *trivial*, PS = 0–0.50; *small*, PS = 0.50–0.56; *moderate*, PS = 0.56–0.71; and *large*, PS > 0.71. The interpretation of the correlation results followed this categorization: r < 0.1, *trivial*; r = 0.1–0.3, *small*; r = 0.3–0.5, *moderate*; r = 0.5–0.7, *large*; r = 0.7–0.9, *very large*; and r > 0.9, *nearly perfect* [45]. The ICC was interpreted according to previously established guidelines as *low* for values below 0.50, *moderate* for values between 0.50 and 0.75, *good* for values between 0.75 and 0.90, and *excellent* for values above 0.90 [46]. The data were analyzed using the Statistical Package for Social Sciences (SPSS version 28, IBM Corporation, Armonk, NY, USA). Statistical significance was set at *p* ≤ 0.05.

## 3. Results

Table 1 shows the descriptive results and reliability values of each of the variables analyzed in the EFEPD-1.0 battery tests. Both the ICC (ICC = 0.65–0.98) and the r/Rho (r/Rho = 0.67–0.98) obtained between R1 and R2 in all the variables analyzed were *moderate–almost perfect*. In all the variables, except for the CMJ_Right_, OLS_Left_, and OLS_Right_, the CVs of the values between R1 and R2 were less than 11.36 ± 0.19%. However, significant differences (*p* < 0.05; r_b_ = −0.85–0.70, *large*) were observed between the R1 and R2 values in the HG_Right,_ HG_Left,_ TUG, 30CTS, 0–10 m, 0–20 m, 10–20 m, COD505, OLS_Left_, and OLS_Right_.

Table 2 shows the results obtained by the participants in the EFEPD-1.0 battery differentiated according to sex. The group of men obtained better results compared to the group of women in the CMJ_Left_ (*p* = 0.03; ES = 0.77, *moderate*), in the HG_Right_ (*p* = 0.000; PS = 0.09, *trivial*), and in the HG_Left_ (*p* = 0.000; PS = 0.09, *trivial*). However, the group of women obtained better values in the OLS_Right_ test (*p* = 0.017; PS = 0.26, *trivial*) and higher values in the HRmax (*p* = 0.038; ES = 0.75, *moderate*), in the HRavg (*p* = 0.039; ES = 0.75, *moderate*), in the Pmin_rel_ (*p* = 0.000; ES = 1.39, *large*), in the Pavg_rel_ (*p* = 0.010; PS = 0.01, *trivial*), and in the Pmin_abs_ (*p* = 0.004; ES = 1.07, *large*), compared to the group of men. In the rest of the variables analyzed, no significant differences were observed between the two groups.

Regarding the correlations between the different tests, the results of the present study showed a *moderate* to *almost perfect* correlation between the acceleration tests, the jump tests, and the TUG (Rho = −0.772–0.880; *p* ≤ 0.001). Figure 1 shows the results of the correlations between the CMJ and the TUG (Figure 1A), the COD505 and the TUG (Figure 1B), and the CMJ and the COD505 (Figure 1C). On the other hand, the results of the HG_Left_, HG_Right_, and OLS_Left_ tests showed significant correlations with the results of the acceleration and jump tests (Rho = −0.543–0.543; *p* ≤ 0.05). However, the OLS_Right_ did not show statistically significant correlations with the jump tests.

Regarding the correlations between the variables recorded in the 6MWT and the rest of the tests (Figure 2), the results showed a *high–very high* correlation between the distance covered (both Dist. and Dist.Stryd) and the acceleration tests, the jump tests (Figure 2A), the TUG test (Figure 2B), and the balance tests (r/Rho = −0.862 to 0.763; *p* ≤ 0.01). No significant correlation was observed between the RPE and the rest of the tests.

In reference to the kinetic and kinematic parameters obtained in the 6MWT, stride length and pace were the kinematic variables that showed the greatest correlation with the acceleration tests, jump tests, and TUG (r/Rho= 0.627–0.896, *p* ≤ 0.01, and *high–very high*). Figure 3 shows the results of the correlations between the stride length and acceleration in 20 m (Figure 3A), CMJ (Figure 3B), COD505 (Figure 3C), and TUG (Figure 3D). On the other hand, the cadence in the 6MWT correlated significantly and *moderately high* (r/Rho= −0.569–0.421; *p* ≤ 0.05) with the acceleration, jump, and TUG variables. Likewise, Pmax_abs_ and Pavg_abs_ were the kinetic variables that obtained the highest correlation (r/Rho = −0.610–0.482, *p* ≤ 0.01, and *moderate–high*) with most of the tests. Finally, the Pmin_rel_ and Pmin_abs_ showed significant but *low–moderate* correlations with some of the acceleration, TUG, and balance tests (r/Rho= −0.431–0.441; *p* ≤ 0.05).

## 4. Discussion

The primary objective of this study was to analyze the validity and reliability of a test battery designed to assess functional capacity in older adults with disabilities or functional limitations. Despite numerous studies that have addressed the analysis of PF in individuals with disabilities, there is a notable lack of standardized tests specifically designed to assess their PF [18]. The EFEPD-1.0 battery demonstrated reliability ranging from *moderate* to *almost perfect* in most of the tests. However, the differences found between the first and second repetitions (R1 and R2) in some tests suggest the need for adequate familiarization of the subjects with the test protocols beforehand. Additionally, significant differences were observed between men and women in most tests of the EFEPD-1.0 battery, highlighting the discriminatory capability of this test battery. Lastly, significant correlations were observed between the various tests of the battery. These findings support the idea that the EFEPD-1.0 battery can be a tool to assess the functional capacity of adults with disabilities.

While the results of the present study highlight the validity and reliability of the EFEPD-1.0 battery, it is essential to acknowledge the limitations inherent to the diversity of the analyzed sample. Although the battery has been designed to be applicable to a wide range of individuals with disabilities, the results suggest the need for specific modifications based on the participants’ individual capacities. For instance, alternative tests or adapted versions may be necessary for individuals who are unable to perform certain evaluations due to physical or cognitive restrictions.

Various studies have shown that most research on the functional capacity of individuals with disabilities has focused on the execution of a series of tests and on the description of the PF data obtained, without having conducted a prior evaluation of validity and reliability [19,37]. Moreover, the use of various test batteries has created difficulties in the interpretation and comparison of data obtained in different studies. Many of these batteries are adaptations of tests intended for non-disabled individuals, with limited evaluation in terms of validity and reliability [19], specifically for individuals with disabilities. Due to these limitations, some authors have recently emphasized the urgent need for a battery specifically designed for adult individuals with disabilities that is valid and reliable [21,22]. In the present study, the tests of the EFEPD-1.0 battery showed good reliability values between repetitions (ICC > 0.81), although in the case of the OLS test, the data obtained were somewhat lower (ICC = 0.65–0.71). Generally, the reliability results obtained were similar [30] or even better [21,24,47] than in previous studies with tests or trials of a similar nature to those included in the EFEPD-1.0 battery. Although the intraclass correlation coefficients between the first and second repetition (R1 and R2) were good in the HG, TUG, 30CTS, sprint COD505, and OLS tests, significant differences in values obtained by participants were observed. These differences between R1 and R2 results may be due to the need for greater familiarization with the protocols by the participants. Although prior familiarization was conducted, adults with disabilities participating in this study might require a more detailed explanation and a more profound and thorough experience of the tests. The differences between R1 and R2 could also be attributed to problems in understanding and functional execution of the tests due to their disability [19]. As suggested by Cabeza-Ruiz et al. [21], more thorough and complete familiarization could reduce these differences and thus improve the reliability values of the EFEPD-1.0 battery.

The differentiated analysis by sex in functional tests for individuals with disabilities has been described as crucial due to inherent variations in physical capacity and responses to rehabilitation between men and women [48]. These differences can significantly influence the assessment and the design of personalized intervention programs [49]. Previous studies indicate that there is notable heterogeneity in the functional profile and motor performance between men and women with disabilities [50], which highlights the importance of adopting differentiated approaches to enhance the quality of life and well-being [51]. In the present study, it was observed that men with disabilities achieved better results in the CMJ_Left_ and in both HG evaluations (neuromuscular capacity measurements), while women performed better in the OLS_Right_ (balance). These results are consistent with previous research where significant differences were observed between men and women with disabilities in tests, such as HG and CMJ [52]. Regarding the results obtained in the 6MWT test (cardiovascular capacity), considering that the group of women showed higher heart rates and power without achieving higher distances compared to the group of men, it appears that the group of women requires greater physiological and mechanical effort to perform the same effective work, denoting lower motor efficiency. These differences, especially in neuromuscular and cardiovascular capacity, could be attributed to several factors. There is a significant difference in muscle mass and body composition between men and women, as occurs in other population groups [53], which affects the ability to generate force and cardiovascular performance. Furthermore, the lower neuromuscular and cardiovascular capacity of women may be due to women with disabilities possibly presenting greater functional deterioration compared to men [52], directly affecting physical performance, in this case neuromuscular and cardiovascular capacity. In this regard, the current battery shows a significant capacity to discriminate between men and women with disabilities, based on the differences observed in various functional capacity parameters. Considering the results, the implementation of physical exercise programs for the development and/or maintenance of physical fitness and functionality related to health from a gender perspective may be highly relevant. In this regard, emphasis on improving strength and cardiovascular capacity could be considered for women with disabilities, as decreased functional capacity can negatively affect their quality of life [9].

Understanding the correlations between different tests can be fundamental to assess how various capacities may influence each other in individuals with disabilities. This understanding not only allows for the identification of areas of strength and weakness in individuals but also can contribute to the creation of more effective intervention strategies. Additionally, in cases where there is limited time to administer the tests, the most representative ones can be selected, as they indirectly provide information about the other tests. In the present study, significant correlations (from *moderate* to *almost perfect*) were observed between the acceleration tests, the jump tests, and the TUG, as well as between the HG_Left_, HG_Right_, and OLS_Left_ with the results of the acceleration and jump tests. Along the same line, previous research observed significant correlations between variants of the TUG and the OLS in individuals with disabilities [54], between linear speed tests (sprint—10 m) and change of direction (COD505) in adults with cerebral palsy [16], and between grip strength and different measures such as force, speed, and power of the jump in older adults [55]. Possibly, the functional tests included in the EFEPD-1.0 battery are interconnected due to the similarity in the capacities they assess (physical or motor functionality). Many of these tests require neuromuscular, coordination, and balance capacities, which are manifested jointly and are interrelated, affecting the final performance in the different evaluations. For example, explosive strength and muscular power are crucial both for the jump tests and for the speed tests [56]. Similarly, the ability to generate force quickly, necessary for good performance in the sprint and vertical jump, may explain the correlations observed between tests like the 0–20, CMJ, SBJ, and TUG. Another possible explanation is that the different functional capacities have a similar evolution in this population. In older adults with disabilities, good strength capacity may be related to other functional capacities due to a concurrent development of these skills over time. This means that individuals with high strength capacity may also show good performance in other functional areas due to similar adaptation in these capacities [55]. The presence of significant correlations between the tests has important implications for the validity of the battery. These associations suggest that the battery consistently and coherently measures the functional capacities of older adults with disabilities. The validity of a test can be evaluated, in part, by the correlation between different tests that assess related aspects of functional capacity [57]. If the tests present adequate correlations among themselves, it can be an indication that they are measuring similar components, an aspect that reinforces the convergent validity of the battery [58].

Although the 6MWT has been widely used in the scientific literature to measure cardiovascular capacity in various populations, including individuals with obesity [59], older adults [60], and individuals with various disabilities [36], so far, few studies [61] have analyzed whether other capacities could influence performance in the 6MWT in individuals with disabilities. The results obtained in the present study showed *large* and *very large* correlations between the distance covered in the 6MWT and the acceleration tests, the jump tests, the TUG, and the balance tests and *moderate* and *large* correlations between the stride length in the 6MWT and the 20 m acceleration, the CMJ, the COD505, and the TUG. These results are consistent with previous research in which a significant correlation was observed between the distance covered in a walking test and the TUG (r = −0.59 and *p* < 0.05; Pedrosa and Holanda [62]), between the HG and the distance covered in the 6MWT in patients with COPD (r = 0.56 and *p* < 0.001; Kovařík et al. [61]), and in older adults (r = 0.5–0.6 and *p* < 0.05; Reis et al. [63]). These results indicate that functionality in the 6MWT is closely associated with the acceleration capacity, the strength of the lower limbs, and the results obtained in the TUG. Consequently, an effective strategy to improve cardiovascular capacity in individuals with disabilities might focus on strengthening these specific capacities. This improvement in acceleration capacity and muscle strength in the lower limbs, along with optimization of the TUG times, could facilitate more efficient displacement over long distances, which in turn would have a positive impact on daily functionality and the quality of life of these individuals. This approach suggests that intervention in multiple functional domains may be key to achieving substantial improvements in the overall health and well-being of individuals with disabilities. Finally, the potential value of utilizing machine learning algorithms for the prediction of states and the enhancement of functional capabilities in individuals with disabilities is a promising avenue for future research, given the efficacy demonstrated by such methods in predicting user states and enhancing functional abilities in people with disabilities [64].

## 5. Conclusions

The present study aimed to design and evaluate the validity and reliability of a functional test battery for individuals with disabilities or functional limitations. The results demonstrate that the EFEPD-1.0 battery is a valid and reliable tool for assessing functional capacity in this population. Convergent validity was evidenced through significant correlations between tests (speed and jump), indicating that these tests assess related capacities or similar constructs. On the other hand, the analysis of discriminant validity revealed significant differences in several tests, demonstrating the battery’s ability to discriminate between men and women. Regarding reliability, most tests showed high consistency, although some observed differences between repetitions indicate the need for greater familiarization with the protocols before their application. On the other hand, an effective strategy to improve functionality in individuals with disabilities could focus on strength training. Nonetheless, the sample, despite its heterogeneity, was modest in size and lacked representation of all possible disabilities, ages, and ethnic groups. Consequently, further studies focused on analyzing discriminant validity between different subgroups would be beneficial, as well as to further refine the battery and improve its applicability and reliability.

## Figures and Tables

**Figure 1 sensors-25-01813-f001:**
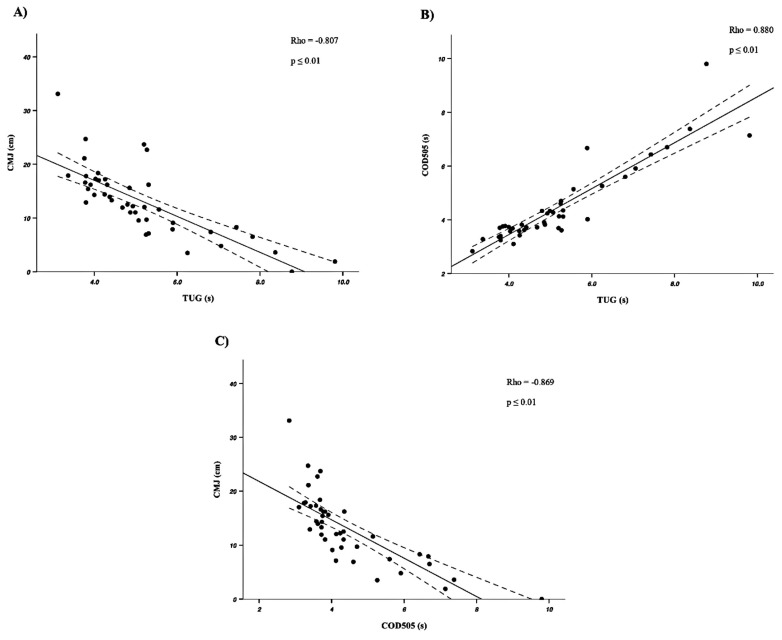
Correlation results between the CMJ test and the TUG (**A**), the COD505 and the TUG (**B**), and the CMJ and the COD505 (**C**).

**Figure 2 sensors-25-01813-f002:**
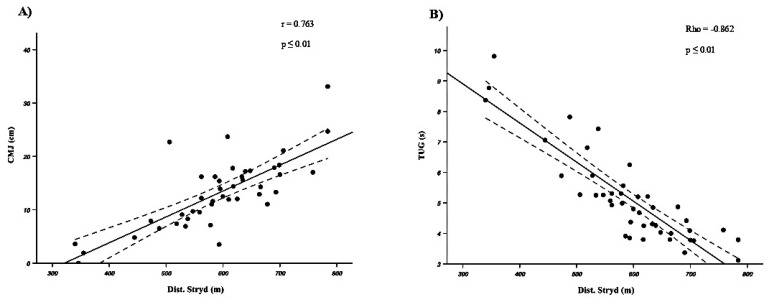
Correlation between the distance covered on the 6MWT and the CMJ (**A**) or the TUG (**B**).

**Figure 3 sensors-25-01813-f003:**
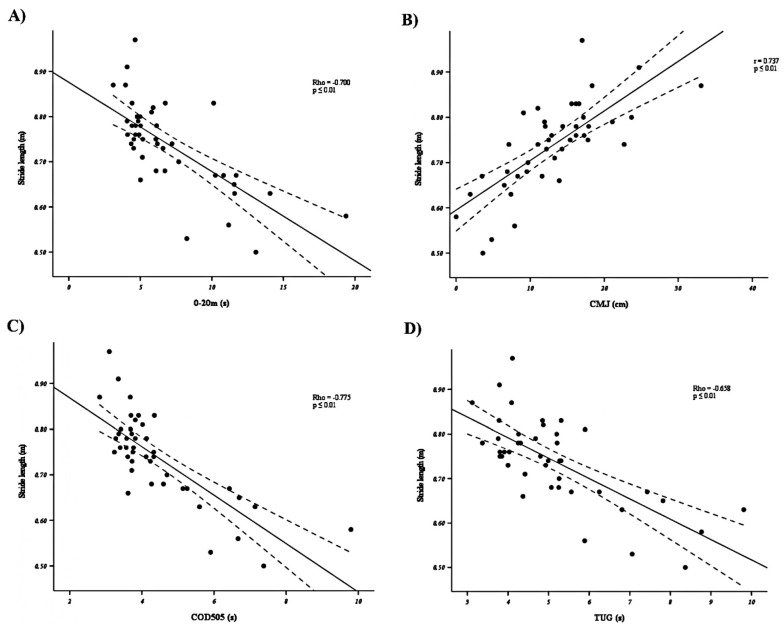
Correlation between stride length and acceleration in 20 m (**A**), CMJ (**B**), COD505 (**C**), and TUG (**D**).

**Table 1 sensors-25-01813-t001:** Descriptive results and test–retest reliability of the tests of the Functionality Evaluation in Population with Disabilities (EFEPD-1.0) battery.

		Best Record	R1	R2	CV	Student-T/Wilcoxon *p* Value	ES d/r_b_	ICC	r/Rho
**Neuromuscular functionality**	CMJ (cm)	13.12 ± 6.52	12.37 ± 6.81	12.71 ± 6.25	11.59% ± 0.23%	*0.16*	*−0.22*	0.97 ***	*0.98 ****
	CMJ_Right_ (cm)	5.58 ± 3.71	4.99 ± 3.76	5.21 ± 3.52	20.27% ± 0.31%	0.14	−0.29	0.93 ***	0.94 ***
	CMJ_Left_ (cm)	5.19 ± 3.52	4.70 ± 3.28	4.89 ± 3.30	10.18% ± 0.10%	0.6	−0.1	0.93 ***	0.96 ***
	SBJ (cm)	0.92 ± 0.41	0.89 ± 0.39	0.87 ± 0.39	11.36% ± 0.19%	*0.46*	*0.11*	0.92 ***	*0.92 ****
	HG_Right_ (kg)	29.57 ± 9.62	26.97 ± 8.20	29.63 ± 9.44	6.46% ± 0.04%	<0.001 ***	−0.58	0.91 ***	0.97 ***
	HG_Left_ (kg)	27.61 ± 8.66	26.14 ± 8.44	27.27 ± 8.35	4.79% ± 0.04%	<0.001 ***	−0.85	0.96 ***	0.97 ***
**Functionality in combined actions**	TUG (s)	5.15 ± 1.51	5.65 ± 2.02	5.34 ± 2.04	6.87% ± 0.07%	<0.001 ***	0.65	0.81 ***	0.96 ***
	30CTS (n°)	14.79 ± 4.08	13.76 ± 4.15	14.39 ± 3.66	7.20% ± 0.07%	<0.001 ***	0.7	0.83 ***	0.93 ***
**Functionality in acceleration**	0–10 m (s)	3.59 ± 1.62	3.79 ± 1.83	3.63 ± 1.64	4.61% ± 0.05%	0.01 **	0.49	0.96 ***	0.95 ***
	0–20 m (s)	6.94 ± 3.43	7.30 ± 3.68	7.00 ± 3.49	3.66% ± 0.03%	<0.001 ***	0.6	0.98 ***	0.99 ***
	10–20 m (s)	3.33 ± 1.81	3.51 ± 1.88	3.36 ± 1.86	4.67% ± 0.05%	<0.001 ***	0.57	0.98 ***	0.98 ***
	COD505 (s)	4.43 ± 1.41	4.68 ± 1.68	4.62 ± 1.67	5.37% ± 0.06%	0.04 *	0.36	0.90 ***	0.91 ***
**Functionality in balance**	OLS_Right_ (s)	15.52 ± 6.70	13.06 ± 7.60	14.81 ± 7.26	23.28% ± 0.33%	0.05 *	−0.63	0.71 ***	0.73 ***
	OLS_Left_ (s)	15.95 ± 6.42	12.96 ± 7.21	15.48 ± 6.96	23.27% ± 0.32%	0.01 **	−0.46	0.65 ***	0.67 ***
**Cardiovascular functionality**	Dist. (m)	564.91 ± 101.64							
	HRmax (pp/m)	137.55 ± 24.68							
	HRavg (pp/m)	123.76 ± 24.68							
	RPEmus	3.46 ± 2.10							
	RPEres	3.00 ± 1.41							
	Ttimp (C°)	36.26 ± 0.63							
	Pmax_rel_ (W/kg)	2.12 ± 0.85							
	Pmin_rel_ (W/kg)	1.33 ± 0.57							
	Pavg_rel_ (W/kg)	1.80 ± 0.74							
	Pmax_abs_ (W)	147.07 ± 37.88							
	Pmin_abs_ (W)	92.44 ± 29.94							
	Pavg_abs_ (W)	123.90 ± 32.29							
	Cadence (p/min)	131.77 ± 13.40							
	Stride length (m)	0.73 ± 0.10							
	Pace (min:s/km)	10:55 ± 3:01							
	GCT (ms)	708.67 ± 308.61							
	Dist.Stryd (m)	592.163 ± 102.64							

Legend: R1 = first record; R2 = second record; CV = coefficient of variation; ES = effect size; d = Cohen’s d; r_b_ = rank-biserial correlation coefficient; ICC = intraclass correlation coefficient; r = Pearson’s correlation; Rho = Spearman’s correlation; CMJ = Countermovement Jump; SBJ = Standing Broad Jump; HG = Hand Grip; TUG = Time Up and Go; 30CTS = 30 s Chair Stand; COD505 = change of direction 505; OLS = One-Leg Stance; Dist. = distance covered by lap count; HRmax = maximum heart rate; HRavg = average heart rate; RPEmus = Rate of Perceived Exertion muscular; RPEres = Rate of Perceived Exertion respiratory; Ttimp = tympanic temperature; Pmax_rel_ = absolute maximum power; Pmin_rel_ = relative minimum power; Pavg_rel_ = relative average power; Pmax_abs_ = absolute maximum power; Pmin_abs_ = absolute minimum power; Pavg_abs_ = absolute average power; GCT = ground contact time; Dist.Stryd = distance covered recorded by Stryd; Italics = parametric; * *p* ≤ 0.05; ** *p* ≤ 0.01; and *** *p* ≤ 0.001.

**Table 2 sensors-25-01813-t002:** Differences between the group of men and women in each of the tests of the Functionality Evaluation of the Population with Disabilities (EFEPD-1.0) battery.

		Men	Women	CV	Student-T/U-Mann *p* Value	ES d/PS
**Neuromuscular functionality**	CMJ (cm)	16.22 ± 8.02	12.06 ± 5.67	20.77%	*0.07*	*0.66*
	CMJ_Right_ (cm)	6.76 ± 4.36	5.18 ± 3.44	18.65%	*0.23*	*0.43*
	CMJ_Left_ (cm)	7.12 ± 3.80	4.53 ± 3.21	31.43%	*0.03 **	*0.77*
	SBJ (cm)	106.78 ± 51.03	86.62 ± 36.11	14.74%	*0.16*	*0.50*
	HG_Right_ (kg)	41.56 ± 8.91	25.45 ± 5.59	34.37%	<0.001 ***	0.09
	HG_Left_ (kg)	38.94 ± 8.49	23.71 ± 4.12	34.01%	<0.001 ***	0.09
**Functionality in combined actions**	TUG (s)	5.11 ± 1.32	5.17 ± 1.58	0.81%	0.80	0.47
	30CTS (n°)	15.09 ± 5.28	14.69 ± 3.67	1.92%	*0.78*	*0.10*
**Functionality in acceleration**	0–10 m (s)	3.21 ± 1.12	3.72 ± 1.75	10.51%	0.31	0.39
	0–20 m (s)	6.06 ± 2.41	7.25 ± 3.70	12.57%	0.30	0.39
	10–20 m (s)	2.85 ± 1.30	3.50 ± 1.94	14.35%	0.30	0.39
	COD505 (s)	4.14 ± 1.10	4.53 ± 1.50	6.48%	0.42	0.42
**Functionality in balance**	OLS_Right_ (s)	11.22 ± 7.21	17.00 ± 5.94	28.96%	<0.01 **	0.26
	OLS_Left_ (s)	13.58 ± 8.26	16.76 ± 5.58	14.82%	0.20	0.39
**Cardiovascular functionality**	Dist. (m)	563.43 ± 111.08	565.42 ± 100.08	0.25%	0.78	0.47
	HRmax (pp/m)	124.36 ± 26.84	142.09 ± 22.58	9.41%	*0.04 **	*0.75*
	HRavg (pp/m)	111.54 ± 26.74	127.96 ± 20.23	9.70%	*0.04 **	*0.75*
	RPEmus	4.27 ± 2.49	3.19 ± 1.92	20.57%	0.12	0.68
	RPEres	2.64 ± 1.28	3.12 ± 1.45	11.99%	0.35	0.41
	Ttimp (C°)	36.20 ± 0.71	36.28 ± 0.61	0.14%	0.71	0.56
	Pmax_rel_ (W/kg)	1.67 ± 0.48	2.28 ± 0.89	21.87%	0.08	0.32
	Pmin_rel_ (W/kg)	0.82 ± 0.23	1.50 ± 0.55	41.67%	*<0.001 ****	*1.39*
	Pavg_rel_ (W/kg)	1.32 ± 0.38	1.96 ± 0.76	27.73%	0.01 **	0.01
	Pmax_abs_ (W)	141.51 ± 32.19	148.99 ± 39.94	3.64%	*0.58*	*0.20*
	Pmin_abs_ (W)	70.71 ± 18.93	99.92 ± 29.56	24.21%	*<0.01 ***	*1.07*
	Pavg_abs_ (W)	111.62 ± 25.45	128.12 ± 33.64	9.73%	*0.15*	*0.52*
	Cadence (p/min)	127.73 ± 12.35	133.16 ± 13.66	2.94%	0.18	0.41
	Stride length (m)	0.78 ± 0.11	0.72 ± 0.08	8.44%	*0.06*	*0.58*
	Pace (min:s/km)	10:18 ± 1:50	11:08 ± 3:20	5.44%	0.94	0.49
	GCT (ms)	793.83 ± 293.36	679.40 ± 312.76	10.98%	0.26	0.61
	Dist.Stryd (m)	604.22 ± 106.07	588.01 ± 102.83	1.98%	1.00	0.50

Legend: CV = coefficient of variation; ES = effect size; d = Cohen’s d; PS = probability of superiority; CMJ = Countermovement Jump; SBJ = Standing Broad Jump; HG = Hand Grip; TUG = Time Up and Go; 30CTS = 30 s Chair Stand; COD505 = change of direction 505; OLS = One-Leg Stance; Dist. = distance covered by lap count; HRmax = maximum heart rate; HRavg = average heart rate; RPEmus = Rate of Perceived Exertion muscular; RPEres = Rate of Perceived Exertion respiratory; Ttimp = tympanic temperature; Pmax_rel_ = absolute maximum power; Pmin_rel_ = relative minimum power; Pavg_rel_ = relative average power; Pmax_abs_ = absolute maximum power; Pmin_abs_ = absolute minimum power; Pavg_abs_ = absolute average power; GCT = ground contact time; Dist.Stryd = distance covered recorded by Stryd; Italics = parametric analysis; * *p* ≤ 0.05; ** *p* ≤ 0.01; and *** *p* ≤ 0.001.

## Data Availability

The data supporting the reported results are available upon reasonable request from the corresponding author. Due to privacy restrictions, the data are not publicly available.

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
