# Peer review of "Proposal for a Battery to Evaluate Functional Capacity in Older Adults with Disabilities"

_sensors, 2025, doi:10.3390/s25061813_

Round 1
Reviewer 1 Report
Comments and Suggestions for Authors
In this manuscript, the authors developed a battery of physical tests for assess functionality in people with disabilities. Although these concepts have been reported, this work presents some new point especially functionality with neuromuscular, combined actions, acceleration, balance under the practical conditions. In addition, the technology shows some advantages such as simplicity, reliability and scalability. Before publication, some other minor issues should be addressed:
> 1. This work is based on the wearable electronics, initial work related to flexible sensors should be cited (Composites Part B: Engineering, 2024, 271, 111152; Advanced Fiber Materials, 2024, 6, 1554; Nano Energy, 2024, 132, 110407)
> 2. The neuromuscular functionality is interesting. But why not accurate measure the neuromuscular using EMGs? Authors claim “The neuromuscular functionality was measured by vertical and horizontal jump test using the optical system Opto Jump Next®“. Accuracy of test methods, errors in optical measurements, whether a more snugly fitting flexible sensor has been considered.
> 3. In figure 2 and 3, It seems that the data is more discrete, and whether such data proves the author's point of view.
> 4. What is the actual classification of people? The physical tests is suitable for Asian or African?
> 5. Can a mechanical learning approach be used to predict the user's state and improve the functional capacities in people with disabilities? (Sci. Adv. 2024, 10, eadq9575; Device 2024, 2, 100366)
Comments on the Quality of English Languagegood
Author Response
|
Response to Reviewer Comments
|
|
Point-by-point response to Comments and Suggestions for Authors
Thank you for the review and comments. First of all, we would like to express our gratitude to Reviewer #1 for the time in reviewing our paper and for providing us helpful comments/suggestions to improve the manuscript. We think that your observations have improved the manuscript. We have answered point-by-point in blue color in this document and in the new version of the manuscript. We find your criticism and recommendations positive and very constructive.
|
|
Comments 1: This work is based on the wearable electronics, initial work related to flexible sensors should be cited (Composites Part B: Engineering, 2024, 271, 111152; Advanced Fiber Materials, 2024, 6, 1554; Nano Energy, 2024, 132, 110407).
Response 1: Thank you for your valuable comment. In response, we have incorporated one of the suggested references into the manuscript. Specifically, it has been included in the following sentence: "The EFEPD-1.0 test battery was subdivided into 5 blocks of functionality and following this order: neuromuscular functionality, functionality in combined action, functionality in acceleration, functionality in balance, and cardiovascular functionality, for which flexible sensors were used to record each test." This addition strengthens the context of our study by highlighting the foundational work in flexible sensors and their relevance to wearable electronics. We appreciate your insightful recommendation, which has significantly enriched the quality of our manuscript.
Comments 2: The neuromuscular functionality is interesting. But why not accurate measure the neuromuscular using EMGs? Authors claim “The neuromuscular functionality was measured by vertical and horizontal jump test using the optical system Opto Jump Next®“. Accuracy of test methods, errors in optical measurements, whether a more snugly fitting flexible sensor has been considered.
Response 2: Thank you for your comment. The aim of our test battery is to ensure that it remains easy to use and administer across a variety of subjects and settings. While we acknowledge that EMG provides higher precision in measuring neuromuscular capacity, our choice of using the vertical and horizontal jump tests with the Opto Jump Next® optical system was deliberate. These tests have been widely employed by other researchers in similar contexts and have proven effective for assessing neuromuscular functionality. Additionally, this approach allows for greater accessibility, ease of application, and time optimization, which are critical factors for the practical implementation of the test battery. We appreciate your observation and believe this rationale aligns with the goals of our study.
Comments 3: In figure 2 and 3, It seems that the data is more discrete, and whether such data proves the author's point of view.
Response 3: We understand your concern. The apparent dispersion in the data presented in Figures 2 and 3 might be attributed to the variability among the study participants. However, the values of “r” and “Rho” are high, which supports the strength of the relationships analyzed. For this reason, we decided to include these figures as a graphical representation of the data, which we believe provides valuable insight into the study's findings. Thank you for pointing this out.
Comments 4: What is the actual classification of people? The physical tests is suitable for Asian or African? Response 4: The present study was conducted with European subjects. We believe that the tests included in the battery are applicable to different populations or ethnic groups, as we do not consider race to be a limitation for performing the assessments incorporated into the battery. However, we have included the following statement in the conclusions section of the manuscript to address this concern: "Nonetheless, the sample, despite its heterogeneity, was modest in size and lacked representation of all possible disabilities, ages, and ethnic groups. Consequently, further studies focused on analysing discriminant validity between different subgroups would be beneficial, as well as to further refine the battery and improve its applicability and reliability." We appreciate your comment, which allowed us to better address the limitations and potential future directions of this research.
Comments 5: Can a mechanical learning approach be used to predict the user's state and improve the functional capacities in people with disabilities? (Sci. Adv. 2024, 10, eadq9575; Device 2024, 2, 100366) |
|
Response 5: Thank you for your insightful comment. In response, we have included the following statement in the manuscript: "Finally, the potential value of utilising machine learning algorithms for the prediction of states and the enhancement of functional capabilities in individuals with disabilities is a promising avenue for future research, given the efficacy demonstrated by such methods in predicting user states and enhancing functional abilities in people with disabilities." We believe this addition highlights an important direction for future exploration and aligns with the advancements mentioned in the references provided. Your suggestion has enriched the scope of our discussion, and we appreciate your contribution. |
|
4. Response to Comments on the Quality of English Language |
|
Point 1: Good |
|
Response 1: Thank you for your positive feedback on the quality of the English language in the manuscript. We appreciate your acknowledgment and have strived to maintain clarity and precision throughout the text. |
Reviewer 2 Report
Comments and Suggestions for Authors
This is an interesting study attempting to validate a series of motility tests in people with disabilities.
The manuscript is well written.
As the authors describe, each of the tests, included in the proposed battery EFEPD-1.0, may be useful in assessing aspects of functionality. Each test has been studied in wide populations, and normative values, cut-offs or goals may apply. Even the duration or itinerary of the tests has been decided accordingly.
As already mentioned in the manuscript, different populations do not respond uniformly to the tests. This may be due to different reasons. Therefore, before making conclusions and setting standards, it is crucial to address whether a test could be realized. Understanding directions is a major component, thus modified versions are necessary. Accordingly, modifications are needed and applied when a person is not able to stand, or even sit, or when a walking aid (cane or frame) is used. From this point of view, I find not appropriate to make conclusions when the sample is diverse. Furthermore, to talk about validation.
It could be useful to comment on the performance of different groups. Yet, the sample size, and sub-groups size is very small. Overall, both the introductory and discussion parts should be revised, according to the literature on the tools and sub-populations in study.
L124 Difficult to understand the meaning.
Author Response
|
Response to Reviewer Comments
|
|
Point-by-point response to Comments and Suggestions for Authors
|
Thank you for the review and comments. First of all, we would like to express our gratitude to Reviewer #2 for the time in reviewing our paper and for providing us helpful comments/suggestions to improve the manuscript. We think that your observations have improved the manuscript. We have answered point-by-point in blue color in this document and in the new version of the manuscript. We find your criticism and recommendations positive and very constructive.
Comment 1: As already mentioned in the manuscript, different populations do not respond uniformly to the tests. This may be due to different reasons. Therefore, before making conclusions and setting standards, it is crucial to address whether a test could be realized. Understanding directions is a major component, thus modified versions are necessary. Accordingly, modifications are needed and applied when a person is not able to stand, or even sit, or when a walking aid (cane or frame) is used. From this point of view, I find not appropriate to make conclusions when the sample is diverse. Furthermore, to talk about validation.
Response 1: Thank you for your insightful comment. As mentioned in the manuscript, we recognize that different populations may respond differently to the tests due to various factors. This variability highlights the importance of tailoring assessments to the specific abilities and conditions of the participants. Modifications to the tests are indeed necessary and have been considered, especially for individuals who cannot stand, sit, or require assistive devices such as canes or frames.
It is important to note that this study is part of a larger project, within which other alternatives or tests have been evaluated for individuals with the mentioned limitations. Consequently, future studies could update and/or complement the current battery with different alternatives or tests that better address these limitations.
Comment 2: It could be useful to comment on the performance of different groups. Yet, the sample size, and sub-groups size is very small. Overall, both the introductory and discussion parts should be revised, according to the literature on the tools and sub-populations in study.
Response 2: Thank you for your insightful comment. In response, we have revised the discussion and conclusions sections of the manuscript to address your observations. We appreciate your suggestion, which has helped to strengthen the manuscript. Please let us know if there are any additional areas that require further attention.
Comment 3: L124 Difficult to understand the meaning.
Response 3: Thank you for your comment. We have corrected the error on line 124.
Round 2
Reviewer 2 Report
Comments and Suggestions for Authors
The authors provided a revised version of their manuscript.
They have included some comments regarding the suitability of their methodology to address functionality.
However, the inherent drawback of a very small and diverse study group still remains.
It is unfortunate, because the statistical analysis is exhausting and detailed, and it would definitely benefit from a solid basic and coherent approach.
